# Case Report of Tetrodotoxin Poisoning from *Lagocephalus sceleratus* in Lebanon

**DOI:** 10.3390/ijerph192214648

**Published:** 2022-11-08

**Authors:** Suad Al-Sulaimani, Nicholas Vincent Titelbaum, Ricardo El Ward, Tharwat El Zahran, Sana Chalhoub, Ziad Kazzi

**Affiliations:** 1Division of Medical Toxicology, Department of Emergency Medicine, Emory University School of Medicine, Atlanta, GA 30307, USA; 2Georgia Poison Center, Atlanta, GA 30303, USA; 3Department of Internal Medicine, Faculty of Medical Sciences, Rafic Hariri University Campus, Lebanese University, Hadath 6573, Lebanon; 4Department of Emergency Medicine, Faculty of Medicine, American University of Beirut Medical Center, Beirut 1107 2020, Lebanon

**Keywords:** tetrodotoxin, pufferfish, neurotoxicity, ingestion

## Abstract

The Indo-Pacific pufferfish *Lagocephalus sceleratus* is a tetrodotoxin-containing species believed to have entered the Mediterranean Sea through the Suez Canal. Tetrodotoxin (TTX) is primarily found in the liver, intestine, and ovaries of *L. sceleratus*. We report a case of a patient with TTX poisoning from *L. sceleratus* consumption in Lebanon. History of ingestion, clinical presentation, and exam findings were obtained during phone-based consultation with the patient and intensive care physician. A 46-year-old male presented to a hospital in Lebanon with perioral and extremity numbness as well as dizziness 1 h after ingestion of an *L. sceleratus* fish. He had caught and prepared the fish himself and had eaten a skinless piece of flesh. Over the following 6 h he also developed ataxia and generalized body numbness. His treatment included systemic hydrocortisone, antihistamine, activated charcoal, and fluids. He was admitted to the intensive care unit, where he developed self-limited, stable sinus bradycardia. He was discharged home on hospital day 5 with residual lightheadedness that improved over several days. This is one of the first reported cases of tetrodotoxin poisoning due to *L. sceleratus* in Lebanon. Public awareness regarding the toxicity of this species after any ingestion is essential to prevent toxicity and death.

## 1. Introduction

Tetrodotoxin (TTX) is a neurotoxin that is found in a variety of animals including pufferfish, blue-ringed octopuses, several shellfish, and some amphibians. TTX is an odorless, tasteless, low molecular weight, heat-stable and non-protein bound potent neurotoxin that was first identified in selected species of pufferfish of the family Tetraodontidae [1]. TTX causes toxicity by blocking fast voltage-gated sodium channels leading to gastrointestinal, neurological, and cardiac clinical manifestations [2]. The onset and severity of TTX toxicity are time- and dose-dependent. Clinical manifestations range from mild, non-specific complaints to muscular paralysis, respiratory failure, and death [3]. A review of TTX incidence and mortality in Japan found that between 1995 and 2010, incidence ranged from 20 to 44 cases per year, with a mortality rate of 0–13.6% [4]. The minimal lethal dose (MLD) of TTX for a 50 kg adult has been estimated to be 2 mg [4,5]. Treatment is supportive and consists of ensuring adequate ventilation, oxygenation, and hemodynamic stability. Currently, there is no antidote for TTX poisoning [6].

TTX is found in more than 20 pufferfish species. The exact mechanism by which TTX occurs in pufferfish is poorly understood, but previous scientific evidence suggests that some pufferfish species can store and accumulate the toxin through a food chain contaminated by TTX-producing bacteria [7,8,9]. This hypothesis is supported by the finding that cultured pufferfish of several species fed in a controlled environment are nontoxic, while wild specimens of the same species contain TTX [10,11]. The levels of TTX found in pufferfish are difficult to predict and depend on a variety of factors including sex, season, size, and maturity. Levels also vary widely among different organs of the same fish specimen. The European Food Safety Authority (EFSA) has determined the safe level of TTX in food for human consumption to be less than 0.044 µg/g of seafood, with toxicity to result from concentrations above this level [12]. The highest TTX levels for the pufferfish species *Lagocephalus sceleratus* have been found in the gonads, livers, intestines, and skin of female fish, and consumption of these organs is typically implicated in cases of clinically significant TTX poisoning by this species [13].

Historically, TTX intoxications have occurred primarily on the coast of Southeast Asia and the Indo-Pacific Ocean region. The Mediterranean Sea is not the natural habitat of the Indo-Pacific *L. sceleratus* [13,14]. However, this species has been increasingly reported from the Mediterranean region in recent years [13]. This range expansion has been attributed to a phenomenon termed the Lessepsian migration, whereby fish species have entered and colonized the Mediterranean Sea from the Red Sea via the manmade Suez Canal [5,14]. Analysis of several pufferfish species caught in Mersin Bay in the northeastern Mediterranean Sea between 2015–2016 demonstrated unsafe levels of TTX in *L. sceleratus, Lagocephalus spadiceus*, *and Lagocephalus suezensis* [10].

We report a patient with TTX poisoning caused by the consumption of the flesh of *L. sceleratus* in Lebanon. The patient in our case demonstrated toxicity without ingestion of the internal organs or skin of the fish.

## 2. Case Description

An otherwise healthy 46-year-old man, presented to a local hospital in Lebanon with perioral and extremity paresthesia associated with dizziness. His symptoms started 1 h after consuming fish. The fish was identified as the *L. sceleratus* pufferfish species by the patient, who was a recreational fisherman and also owned a fishery (Figure 1). He had caught the fish off a sandy coast in Jounieh, Lebanon. The patient reported that he consumed 500 g of the fish. He cleaned and prepared the fish himself and ate a fried piece of the flesh without its skin.

One hour after consuming the fish, he began to develop dizziness, ataxia, and generalized body numbness. After 6 h, the patient’s symptoms had progressed to severe difficulty with ambulation, and he presented to the emergency department of a local hospital. On arrival he was afebrile, normotensive with a blood pressure of 120/65 mmHg, had a pulse of 75 beats/min, and was saturating 96% on room air. The patient was alert and oriented and exhibited ataxia with an otherwise unremarkable physical exam. The patient was admitted to the intensive care unit and received supportive care. His condition progressed over the subsequent few hours, and he developed bradycardia with a pulse of 40 beats/min with normal blood pressure and good perfusion (Figure 2). The bradycardia resolved spontaneously after a few hours. 

His laboratory parameters remained within normal limits throughout his hospital stay, with initial tests reporting a white blood count of 6.4 10^3^/µL, hemoglobin of 14.6 g/dL, platelets of 326/µL, sodium of 137 mmol/L, potassium of 4.5 mmol/L, calcium of 9.5 mg/dL, magnesium of 2 mg/dL, and computed tomography of his head was normal. His hospital medical treatment included systemic hydrocortisone, an antihistamine, and activated charcoal 1 g/kg given at 10 h post-ingestion. 

The patient also received mannitol 0.5 g/kg and intravenous fluids. Over the next 24 h, the patient’s clinical status began to improve. By day 5 of his hospital stay, his numbness had resolved, but he reported persistent dizziness. He was discharged home with residual lightheadedness that improved over the next few days after discharge. 

## 3. Discussion

TTX toxicity has been associated with consumption of *L. sceleratus* internal organs such as the liver or gonads, but this case demonstrates a patient presenting with signs and symptoms consistent with TTX toxicity after consumption of only the muscular flesh. Although studies have postulated trends regarding TTX concentration by organ and season in *L. sceleratus,* inconsistencies exist and consumption of any part of *L. sceleratus* during any season should be avoided [15]. 

Several recent studies demonstrated the presence of tetrodotoxin in the liver, gonad, intestine, muscle, and skin of all specimens. Average concentrations were lower in the muscle and skin compared with the liver, gonad, and intestine [1,15,16,17]. 

Some studies of *L. sceleratus* in the Mediterranean have found that TTX concentrations in liver, intestine, muscle, and skin are highest in the spring and summer, coinciding with the species’ reproductive period [16]. However, this trend is not consistently observed, as another study of TTX in *L. sceleratus* from Mersin Bay, Turkey found concentrations in most organs to be highest in autumn or winter [13].

The first case of TTX toxicity from *L. sceleratus* in Lebanon occurred in 2004 [18], followed in January 2008 by a second case that was reported in the literature [19]. Both cases developed severe toxicity after consumption of the fish liver. In the first, unreported case, the patient developed hemodynamic and respiratory failure that required mechanical ventilation and vasopressors. The patient was comatose, and his condition improved after hemodialysis. Although he recovered and was discharged from the hospital, no further follow-up was available [18]. In the second case, the patient developed proximal weakness progressing to quadriplegia and dysarthria, ophthalmoplegia, dyspnea, and absent gag reflex. The patient required intubation for 4 days but made a full recovery without neurologic sequelae. The third reported case of TTX toxicity from *L. sceleratus* in Lebanon was published in 2010 [20]. The patient ate the gonads of the fish and presented with perioral tingling followed by dysarthria, quadriplegia, respiratory distress, and hemodynamic failure within 3–4 h. He lost all brainstem reflexes but began to improve after 20 h with full recovery to neurologic baseline within 36 h [20]. Additionally, a 2014 study of *L. sceleratus* population structure and sexual maturity in Lebanon and Syria mentions two cases of human TTX toxicity from pufferfish consumption in Lebanon [21]. 

A case series from Israel of 13 patients admitted after consumption of *L. sceleratus* reported a single case of a patient who ate a plate of musculature and remained asymptomatic, whereas all who ate more than a very small taste of the liver developed signs and symptoms consistent with toxicity [14]. Other reported cases of TTX toxicity from *L. sceleratus* developed symptoms after consuming liver, gonads, or do not specify what part of the fish was consumed [22,23]. Although most cases of TTX toxicity from *L. sceleratus* involve consumption of the organs, our case demonstrates that no part of *L. sceleratus* should be considered safe to eat.

The management of TTX toxicity is primarily supportive with no role for mannitol. Administration of activated charcoal is reasonable in the first hour after ingestion. Administration of anticholinesterases such as edrophonium or neostigmine to inhibit the breakdown of acetylcholine at the neuromuscular junction has been used in some case series with improvement in motor function [3,22,24]. Hemodialysis use in TTX poisoning has been reported in the literature. Unfortunately, there is currently insufficient evidence to support its use. In a recent case series of five patients from Oman, hemodialysis was performed in two intubated patients with TTX exposure confirmed by urine testing [3]. The first patient received continuous venovenous hemofiltration followed by intermittent dialysis and showed dramatic improvement 15 h after starting dialysis. The patient was extubated after 57 h. The second patient showed improvement after the first session of hemodialysis and was extubated a few hours later. The authors propose further study or evaluation of hemodialysis in severe TTX poisoning, as the toxin is small in size and not protein-bound [3].

## 4. Conclusions

Since it was first reported in the Mediterranean Sea in 2004, *L. sceleratus* has rapidly expanded its presence throughout the region, with reports spanning from Lebanon to Turkey, and several specimens even found in the Black Sea. The expansion of TTX-containing fish species such as *L. sceleratus* into the Mediterranean Sea via the Suez Canal poses a unique public health hazard, as fishermen are unfamiliar with the dangers of eating such species and may catch them for sale or personal consumption. Given the potential life-threatening effects of TTX, further efforts are needed to spread awareness regarding the Lessepsian migration phenomenon and that consumption of any part of *L. sceleratus* should be avoided all times of the year. Guidelines for proper management of these intoxications should also be disseminated in Mediterranean countries.

## Figures and Tables

**Figure 1 ijerph-19-14648-f001:**
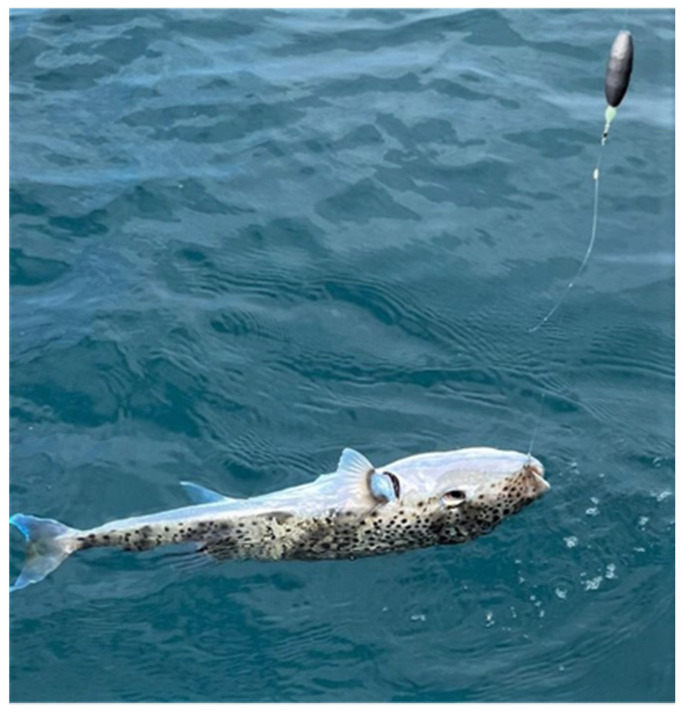
An image of the ingested fish, *Lagocephalus sceleratus*, taken by the patient.

**Figure 2 ijerph-19-14648-f002:**
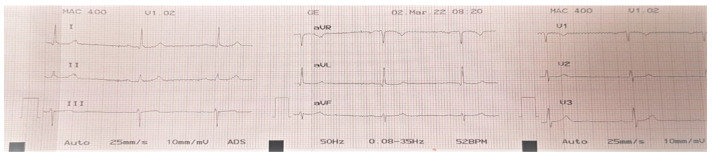
Electrocardiogram of the patient obtained during a period of bradycardia while in the intensive care unit.

## Data Availability

Not applicable.

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
