# Peer review of "Case Report of Tetrodotoxin Poisoning from *Lagocephalus sceleratus* in Lebanon"

_ijerph, 2022, doi:10.3390/ijerph192214648_

Round 1
Reviewer 1 Report
The work assessed concerns a very important aspect of public health. The subject matter undertaken by the authors is directly related to the impact of the conditions of the natural environment on human life. The emerging threats are very often related to toxins.
I am full of appreciation for the authors' work and motivation. Unfortunately, the case report provided by the authors is incomplete. In each scientific work, it is necessary to define: the effect of the work, tools, method of analysis, the conditions in which the research was carried out, the estimated accuracy of the analysis, discussion and conclusions. We must present all these elements against the background of the current state of knowledge.
Some elements in the work are missing or their quality is insufficient. In my opinion, the case report needs to be rewritten in its entirety. Moreover, I would like to emphasize that the topic is important and should be published by the authors, but in a modified form.
Author Response
Thank you for your comments. We would like to request clarification about your statement on the need to define: the effect of the work, tools, methods of analysis, the conditions in which the research was carried out, the estimated accuracy of the analysis and the discussion and conclusions. As this is a case report, we do not have a statistical analysis. The toxin analysis is unavailable in Lebanon and the accuracy of the report is assured by the treating physicians who have interviewed the patient in person. The impact of this report is on public health as it raises awareness of this recently introduced marine species in the Mediterranean Sea. We would be happy to provide additional information as requested by the reviewer, however we are uncertain about the specific recommendation to re-write the manuscript.
Reviewer 2 Report
Dear Author,
This report about Tetrodotoxin poisoning is well-planned and clearly presented. In my opinion, it is helpful in raising public awareness throughout the Mediterranean countries regarding the risk of poisoning via consumption of unconsciously hunted marine animals. Moreover, from a clinical perspective, this report can be guided healthcare professionals in managing Tetrodotoxin poisoning. After minor revisions, it is appropriate to be published in the International Journal of Environmental Research and Public Health. A detailed explanation has been made in the “Comments” section.
COMMENTS:
In the current study, the authors have presented the case of Tetrodotoxin poisoning after the consumption of a pufferfish. Interestingly, the patient was poisoned after consumption of 500 gr of fried flesh without eating the internal organs and skin, where the tetrodotoxin is much more accumulated. In my opinion, this report is well-planned and clearly presented.
However, there are a few specific points that should be addressed.
- The introduction section can be improved. For example, the bacterial origin of Tetrodotoxin in pufferfish species and how the amount of Tetradotoxin can reach toxic levels (via the aquatic food chain?) can be mentioned in detail. The lethal dose of and/or the dose level which is not expected to result in adverse effects for humans can be indicated.
-Line 37, as far as i understand, the given incidences for Japan are annual, therefore relevant part can be changed to "20 to 44 cases per year”.
-Line 205 (ref.15), please provide the exact date.
Author Response
Thank you very much for your feedback and comments. Please find a point-by-point response to each comment below:
Comment 1: The introduction section can be improved. For example, the bacterial origin of Tetrodotoxin in pufferfish species and how the amount of Tetradotoxin can reach toxic levels (via the aquatic food chain?) can be mentioned in detail. The lethal dose of and/or the dose level which is not expected to result in adverse effects for humans can be indicated.
Response to Comment 1: A more detailed discussion regarding the origin of tetrodotoxin has been provided in lines 41-48. The lethal dose is mentioned in lines 37-38. The level which is not expected to result in adverse effects for humans is indicated in lines 49-51.
Comment 2: Line 37, as far as i understand, the given incidences for Japan are annual, therefore relevant part can be changed to "20 to 44 cases per year
Response to Comment 2: This modification has been made.
Comment 3: Line 205 (ref.15), please provide the exact date.
Response to Comment 3: This modification has been made.
Round 2
Reviewer 1 Report
The added corrections to the article do not raise any doubts and improve the quality of the data.